# Calcium-Based Antimicrobial Peptide Compounds Attenuate DNFB-Induced Atopic Dermatitis-Like Skin Lesions via Th-Cells in BALB/c Mice

**DOI:** 10.3390/ijms231911371

**Published:** 2022-09-26

**Authors:** Qingfeng Liu, Mengmeng Li, Na Wang, Chun He, Xian Jiang, Jingyi Li

**Affiliations:** Department of Dermatology and Venereology, West China Hospital of Sichuan University, Chengdu 610041, China

**Keywords:** atopic dermatitis, calcium-based antimicrobial peptides compounds, BALB/c mice, cytokines

## Abstract

Atopic dermatitis (AD) is a chronic and recurrent inflammatory skin disease, characterized by severe itching and recurrent skin lesions. We hypothesized that a novel treatment involving calcium-based antimicrobial peptide compounds (CAPCS), a combination of natural calcium extracted from marine shellfish, and a variety of antimicrobial peptides, may be beneficial for AD. We established a dinitrofluorobenzene (DNFB)-induced AD model in BALB/c mice to test our hypothesis. We observed mouse behavior and conducted histopathological and immunohistochemical analyses on skin lesions before and after CAPCS treatment. We also characterized the changes in the levels of cytokines, inflammatory mediators, and Toll-like receptors (TLRs) in plasma and skin lesions. The results showed that (i) topical application of CAPCS ameliorated AD-like skin lesions and reduced scratching behavior in BALB/c mice; (ii) CAPCS suppressed infiltration of inflammatory cells and inhibited the expression of inflammatory cytokines in AD-like skin lesions; (iii) CAPCS reduced plasma levels of inflammatory cytokines; and (iv) CAPCS inhibited TLR2 and TLR4 protein expression in skin lesions. Topical application of CAPCS exhibits a therapeutic effect on AD by inhibiting inflammatory immune responses via recruiting helper T cells and engaging the TLR2 and TLR4 signaling pathways. Therefore, CAPCS may be useful for the treatment of AD.

## 1. Introduction

Atopic dermatitis (AD) is a chronic and recurrent inflammatory skin disease, characterized by severe itching and recurrent skin lesions [1,2]. The prevalence of AD is approximately 20% in children and ranges between 7% and 14% among adults in Europe and the United States [3]. There were 390 million AD patients worldwide in 2019, and the number was estimated to reach 450 million and 520 million by 2024 and 2030, respectively [4]. For moderate and severe AD, the average course was nearly 10 years, and the overall affected lesion area was more than 30% of the skin. Most of the patients suffering from moderate and severe AD have difficulty sleeping due to unbearable itching, and reducing itching symptoms is the crucial and urgent need for 75.8% of them. More than 10% of them had suicidal tendencies, and 71.2% of them had experienced discrimination [5]. The pathogenesis of AD is not completely clear, and it is generally believed that the immune system is involved [6]. It is reported that various cytokines secreted by CD4^+^ T cell subsets, such as Th1, Th2, Th17, and Th22 cells, participate in the pathogenesis of AD at different stages, and the composition of these cytokines is dependent on age, race, and disease severity [7,8]. The function of the skin barrier can be affected by cytokines via multiple pathways, including downregulated expression of keratinocyte differentiation genes, alterations in skin symbiotic microorganisms, and changes in the structure of the skin barrier [9,10]. Toll-like receptors (TLRs) also play an important role in antimicrobial infection. The expression of TLR2 and TLR4 on the surface of peripheral blood mononuclear cells is decreased in patients with AD, in comparison to healthy volunteers, and the loss of TLR2 and TLR4 signals is believed to induce the Th2 immune response, leading to the development of AD [11,12].

Antimicrobial peptides (AMPs), including human β-defensin (HBD)-1-3, actinomycin, LL-37, RNase7, dermatophytin, and insecticidal, are major components of the innate immune system; they have been reported to have antibacterial and bactericidal activities [13,14]. Furthermore, reduced levels of AMPs in AD patients are hypothesized to facilitate infection and colonization by *Staphylococcus aureus* on the skin [15,16]. In addition, the calcium concentration gradient in the epidermis is reported to play a key role in keratinocyte differentiation, skin barrier formation, and homeostasis [17]. Calcium-based antimicrobial peptide compounds (CAPCS) are a combination of natural active calcium extracted from marine shellfish and a variety of AMPs. The shells for preparation of CAPCS were crushed and placed under vacuum conditions and 600 °C for 3 h, calcinated under mixed gas (40% air and 60% CO) and 1000 °C for 2 h, and then crushed with a micro nano pulverizer to obtain shell nano powders characterized with calcium carbonate porous body containing calcium oxide structure inside. With 10 mg of the prepared AMPs embedded in 1 kg of the shell nano powders by using a core-entrapment technology, the bio-composite antibacterial adsorbent CAPCS were prepared [18]. Theoretically, CAPCS can produce the effects of both calcium concentration gradients and AMP provision. Based on this rationale, we hypothesized that CAPCS may have a therapeutic effect on AD. We tested the hypothesis in an AD mouse model by observing the behavior of the animals and by characterizing AD-like skin lesions before and after CAPCS treatment, through histopathology and immunohistochemistry. We also determined the changes in cytokines, inflammatory mediators, and TLRs in serum and in skin lesions.

## 2. Results

### 2.1. Topical Application of CAPCS Ameliorates DNFB-Induced AD-Like Skin Lesions and Reduces Scratching Behavior in BALB/c Mice

AD-like skin lesions were observed in the DNFB-treated (DT), DNFB with vesicle treated (DVT), DNFB with CAPCS-treated (DCT), and DNFB with desonide-treated (DDT) groups, while no skin lesions were observed in the DNFB non-treated (NT) group. The dermatitis scores were 0 in NT group, 8.33 ± 0.58 in DT group, 7.33 ± 0.72 in DVT group, 1.67 ± 0.38 in DT group, and 1.33 ± 0.58 in DVT group. The scores were significantly lower in the DCT and DDT groups, in comparison to the DT and DVT groups (*p* < 0.05), and the scores were similar between the DCT and DDT groups without statistical significance (Figure 1). The scratching behavior of mice was obviously increased in DVT group (33.00 ± 5.29 times/10 min) than DT groups (5.33 ± 0.58 times/10 min) and reduced significantly in the DCT and DDT groups (0.67 ± 1.15 and 0, respectively), and there were no significant differences observed between DCT and DDT groups (Figure 2).

### 2.2. CAPCS Suppresses Infiltration of Inflammatory Cells and Inhibits Expression of Inflammatory Cytokines in DNFB-Induced AD-Like Skin Lesions of BALB/c Mice

The expression of inflammatory cytokines, including IFN-γ, IL-4, IL-5, IL-13, IL-31, IL-17, and IL-22, was significantly increased in DNFB-induced AD-like dorsal skin lesions of the DT, DVT, DCT, and DDT groups, in comparison to normal dorsal skin in the NT group. Furthermore, cytokine expression levels were significantly decreased in the DCT and DDT groups, in comparison to the DT and DVT groups (*p* < 0.05) (Figure 3). There were no significant differences observed between the DCT and DDT groups and the DT and DVT groups (Figure 1).

### 2.3. CAPCS Suppresses Plasma Levels of Inflammatory Cytokines and the Corresponding Protein Expression Levels in BALB/c Mice

The expression of plasma inflammatory cytokines, including IFN-γ, IL-4, IL-5, IL-13, IL-31, IL-17, and IL-22, was significantly increased in the DT, DVT, DCT, and DDT groups, in comparison to the NT group (*p* < 0.05). Moreover, the levels of cytokine expression were significantly decreased in the DCT and DDT groups, in comparison to the DT and DVT groups (*p* < 0.05) (Figure 4). There were no significant differences observed between the DCT and DDT groups, and the DT and DVT groups (Figure 3).

The mRNA levels of IFN-γ, IL-4, IL-5, IL-13, IL-31, IL-17, and IL-22 were significantly increased in the DT, DVT, DCT, and DDT groups, in comparison to the NT group (*p* < 0.05). Moreover, the mRNA levels were significantly decreased in the DCT and DDT groups, in comparison to the DT and DVT groups (*p* < 0.05) (Figure 5). There were no significant differences observed between the DCT and DDT groups and between the DT and DVT groups (Figure 4).

### 2.4. CAPCS Inhibits TLR2 and TLR4 Protein Expression in BALB/c Mice with AD-Like Clinical Symptoms

TLR2 and TLR4 protein expression was significantly increased in the DT, DVT, DCT, and DDT groups, in comparison to the NT group (*p* < 0.05). Moreover, their expression was significantly decreased in skin lesions of the DCT and DDT groups, in comparison to the DT and DVT groups (*p* < 0.05). There were no significant differences between the DCT and DDT groups and the DT and DVT groups (Figure 5).

## 3. Discussion

In this study, we found that (i) topical application of CAPCS ameliorated DNFB-induced AD-like skin lesions and reduced scratching behavior in BALB/c mice; (ii) CAPCS suppressed the infiltration of inflammatory cells and inhibited the expression of inflammatory cytokines in DNFB-induced AD-like skin lesions; (iii) CAPCS suppressed DNFB-induced serum levels of inflammatory cytokines; and (iv) CAPCS inhibited TLR2 and TLR4 protein expression in the skin lesions of BALB/c mice. Based on these results, we suggest that CAPCS may have a therapeutic effect on AD. To our knowledge, this is the first study to explore the efficacy and mechanism of action of CAPCS in AD treatment, and this study confirmed our proposed hypothesis that CAPCS have a therapeutic effect on AD.

Severe pruritus is a prominent and burdensome symptom of AD, both in humans and in established mammal models of AD [1,19]. Scratching induced by pruritus exacerbates skin lesions and causes sleep disturbances that decrease the quality of life and impair psychosocial well-being. In our study, we found that the application of CAPCS reduced scratching behavior in BALB/c mice with AD-like symptoms and relieved DNFB-induced AD-like skin lesions, and this therapeutic effect was similar to that of desonide, a weakly potent corticosteroid. This finding suggests that CAPCS treatment may be effective for ameliorating AD-associated pruritus.

The pathogenesis of pruritus in AD is complicated and involves various factors, including (i) disrupted epidermal barrier allowing the entry of itch-causing substances, such as mechanical irritation from animal hair or fabrics, bacterial infection, and allergens [20]; (ii) neural pathway with non-histaminergic C fibers and neural hypersensitization [21]; and (iii) endogenous pruritogenic mediators, such as inflammatory factors or cytokines, that stimulate and activate cutaneous pruritus-sensing nerves [1,2].

CAPCS is a compound biological component composed of a variety of AMPs combined with a natural micro-calcium substrate extracted from marine shellfish using embedding technology [18]. AMPs in CAPCS can change the permeability of the bacterial cell membrane to form an opening or channel within the cell membrane, and calcium ions released from the micro-calcium substrate can rapidly enter the bacterial cells. As a large number of calcium ions enter the cells, the concentration of intracellular calcium is rapidly increased, resulting in cell metabolism disorder and intracellular organelles leaking out under the action of both intracellular and extracellular pressure, leading to the loss of the survival ability of the pathogenic bacteria [18]. M. Kazemzadeh-Narbat et al. found that antimicrobial peptides on calcium phosphate-coated titanium could prevent implant-associated infections [22]. Thus, it may reveal a similar mechanism as our study. In the current study, histopathological evaluation of dorsal skin lesions demonstrated that decreased inflammatory cell infiltration was observed in the CAPCS-treated group, suggesting that one of the therapeutic effects of CAPCS for AD could be due to AMPs, which improved skin barrier function and reduced the inflammatory response to relieve pruritus in AD.

According to the reports about toxicity, CAPCS did not induce structural chromosome aberrations in cultured Chinese hamster lung (CHL) cells and did not show potential toxicity to mouse fibroblast L-929 cells in vitro cytotoxicity test. There were no evidence of causing acute systemic toxicity in the mouse, or subacute systemic toxicity in the rats, or skin sensitization in the guinea pig in term of CAPCS, and CAPCS did not irritate the subcutaneous tissue or induce intracutaneous reactivity in rabbits (Unpublished data). These findings suggested that CAPCS was not associated with observed side effects on normal skin.

Inflammatory cytokines play an important role in the pathogenesis of AD. In our study, we found that expression of IFN-γ (Th1 cell cytokine), IL-4, IL-5, IL-13, IL-31 (Th2 cell cytokines), IL-17, and IL-22 (Th17 and Th22 cell cytokines) were increased in both plasma and AD-like skin lesions of DNFB treated mice. These findings consistent with previous literatures, in that the expression levels of multiple cytokines were elevated in skin lesions of patients with AD or established AD mouse models [23,24,25,26,27,28]. Activation of type 2 helper T (Th2) cells has been reported to be involved in the acute phase of AD, and the increased expression of Th2 cell-related cytokines IL-4, IL-13, IL-5, and IL-31 were implicated in its pathophysiology [23]. Transgenic mice with epidermis overexpressing IL-4 and IL-13 exhibited AD-like symptoms, including skin lesions and pruritus [24], and both IL-31 transgenic mice and wild-type mice treated with IL-31 developed hallmark features of AD and scratching behavior [25]. IFN-γ secreted by Th1 cells causes abnormal lipid composition and leads to the destruction of the skin barrier in AD [26]. IL-17 has been reported to trigger Th2 responses, such as IL-4 production in AD progression, and IL-22 has been found to be positively correlated with the severity of AD [27,28]. The new findings in our study were that all of cytokines, including Th1 (IFN-γ), Th2 (IL-4, IL-13, Il-5, and IL-31), Th17 (IL-17), and Th22 (IL-22), were significantly decreased in AD-like skin lesions and plasma after application of CAPCS, and the levels were similar to those in the group treated with desonide. These results further support the notion that the therapeutic effect of CAPCS on AD is similar to that of a weak corticosteroid.

TLRs are important regulators of the immune system, and their single nucleotide polymorphisms are related to the pathogenesis and progression of AD [12,29]. TLR signaling plays a crucial role in host defense against danger signals by producing a variety of cytokines, chemokines, antimicrobial peptides, and costimulatory factors. Furthermore, it has been reported how to make the skin of AD patients more vulnerable to infections, especially the colonization of *S. aureus* in skin lesions [12,29]. In the current study, we found that the expression levels of TLR2 and TLR4 were increased in the AD model of DNFB-induced AD mice and decreased significantly after treatment with CAPCS and desonide. These results suggest that CAPCS inhibits TLR-mediated immune responses in AD, and this inhibitory effect is similar to that of a weak corticosteroid.

Importantly, we found that all experiment groups, including vehicle control, showed therapeutic effects on AD-like lesions. These effects could be due to the discontinuation of DNFB on day 9 and the moisturizing effect of the vehicle. However, mice treated with CAPCS and desonide had a better therapeutic outcome than mice treated with the vehicle alone, including significantly decreased levels of inflammatory cytokines and TLR2 and TLR4 protein expression.

There were several limitations in the current study. First, the observation time was only two weeks, and we had no data about the long-term effects of CAPCS treatment on AD. Second, the mice model of AD-like clinical symptoms was induced by DNFB, and the skin lesions were improved spontaneously with the cessation of DNFB stimulation, which was different from natural process of AD and clinical practice. Third, we did not have a bacteriological examination of the skin lesion, especially *S. aureus*, so we could not explore the antibacterial effect of CAPCS. Fourth, we did not design a group with a cocktail of CAPCS and desonide treatment, which might provide more information on whether the two drugs were under similar mechanisms and help enhance the AD-like lesion healing. In this research, we tend to focus on the therapeutic effect of CAPCS, and desonide treatment was designed as a comparator group, so we thought that the five groups should be sufficient to draw the conclusion. In future studies, we plan to design animal experiments and clinical trials to investigate the short-term (less than three months) and long-term (six months to two years) effects of CAPCS on AD and explore the advanced mechanism of CAPCS treatment, such as antibacterial effect, the signal pathway, or the micro RNA level.

In conclusion, topical application of CAPCS exhibits a therapeutic effect on AD by inhibiting inflammatory immune responses via TLR2 and TLR4 signaling pathways. CAPCS may be useful for the treatment of AD, especially in cases where steroid treatment is not an option.

## 4. Materials and Methods

### 4.1. Animals and Reagents

Seven-week-old specific-pathogen-free (SPF)-grade female BALB/c mice were provided by the Animal Experimental Center of Sichuan University and housed with 10 mice per cage in a standard animal feeding room under SPF conditions. The temperature was maintained at 21 ± 2 °C, with a relative humidity of 55 ± 5%, and a 12 h dark/light cycle was employed. Mice were provided with standard laboratory feed. All animal experiments were conducted in accordance with the regulations of the Administration of Experimental Animals (revised 2017) published by the State Council of the People’s Republic of China and were approved by the Animal Ethics Committee of West China Hospital of Sichuan University (Ethical approval number: 20211203A, approval date: 18 June 2021).

Dinitrofluorobenzene (DNFB) was purchased from Sigma-Aldrich (St. Louis, MO, USA), CAPCS was provided by Shell Party Innovations (SPI) Co., Ltd. (Shenzhen, China), and desonide was purchased from Chongqing Huapont Pharm Co., Ltd. (Chongqing, China). Mouse ELISA kits for IFN-γ (EMC101g.96), IL-4 (EMC003.96), IL-5 (EMC108.96), IL-13 (EMC124.96), IL-17 (EMC008.96), and IL-22 (EMC119.96) were provided by Xinbosheng Biotechnology (Shenzhen, China), and kit for IL-31 (ARG81805) was provided by Arigo Biolaboratories (Shanghai, China). The rabbit monoclonal TLR2 antibody (ab209216) and the mouse monoclonal TLR4 antibody (ab22048) were provided by Abcam Trading Co. (Shanghai, China).

### 4.2. Induction of AD-Like Skin Lesions in Mice and Group Allocation

DNFB was used as a stimulant to induce AD-like clinical symptoms, according to the study reported by Tian et al. [30] with minor modifications. Mice (*n* = 60) were randomly assigned to five groups (*n* = 12 per group): NT group (non-treatment, mice without any treatment), DT group (DNFB-treated AD mice without drug treatment), DVT group (DNFB-treated AD mice plus vehicle treatment), DCT group (DNFB-treated AD mice plus CAPCS treatment), and DDT group (DNFB-treated AD mice plus desonide treatment).

One day before treatment, the mice were shaved on the abdomen and back (approximately 2 cm × 3 cm). Apart from the mice allocated to the NT group, the others were sensitized by smearing 100 μL of 0.5% DNFB on the shaved area of the abdomen. Four days later, 20 μL of 0.2% DNFB was applied to the shaving zone of the abdomen and back once a day for 5 days. Successful establishment of the model included erythema, blisters, and exudation, appearing on the corresponding part of the skin of the mice, some of which were eroded, covered with yellow crusts, and exhibited exudation, swelling, and scabbing in the ears. The mice in the NT group were fed as normal without any treatment, the mice in the DT group were not given any other treatment, and the mice in the DVT, DCT, and DDT groups were treated with repeated application of vehicle, CAPCS, and desonide, respectively, on the AD-like lesions twice a day for 14 days (Figure 6).

### 4.3. Dermatitis Score and Scratching Behavior

The AD-like skin lesions were evaluated macroscopically and scored based on the severity of four symptoms including erythema/oozing, edema, dryness/excoriation, and crust/lichenification. Each lesion was graded into four scores: 0, none, without any sign of erythema/oozing, edema, dryness/excoriation, or crust/lichenification; 1, mild, with mild but noticeable erythema, edema, dryness or crust and without oozing, excoriation, or lichenification; 2, moderate, with obvious erythema and/or mild oozing, obvious edema, noticeable dryness and/or mild excoriation, or obvious curst and/or mild lichenification; and 3, severe, with obvious and severe oozing, edema, excoriation, and lichenification. The dermatitis score in each mouse was determined by the sum of the scores of the four symptoms, and the final score would be 0~12. The evaluation was performed by a researcher blinded to the group allocation of the mice.

The mice in each group were placed in a single cage, and the scratching behavior of the mice was observed every 3 days before and after successful modeling. The scratching times of the mice for 10 min were recorded and counted using a video camera. When the hind paw falls to the ground or is licked, it indicates the end of scratching. Movement of the forelimb and scratching of the face by the hind paw was not counted.

### 4.4. Blood Collection from Cardiac Apex

The mice were anesthetized with the 0.1% of pentobarbital sodium, laid on the back with the limbs fixed by rubberized fabric on a slab, then the thoracic cavity were opened. After locating the cardiac apex, a 1 mL syringe with a 22 gauge needle was inserted to get the blood. The blood was transferred into the tubes with ethylenediaminetetraacetic acid (EDTA) and centrifuged at 2000× *g* for 10 min at 4 °C, and the supernatant was collected.

### 4.5. Histopathology

After completion of all treatments, dorsal skin lesions were removed, fixed with 4% formaldehyde, embedded in paraffin, and sliced. Then, the sections were stained with hematoxylin and eosin (H&E) and inflammatory changes in skin tissues were observed under a light microscope. The number of neutrophils was counted in 10 non-overlapping visual fields at 400× magnification. Pathological changes in skin lesion thickness, inflammatory cell infiltration, CD4^+^ cells, mast cells, and eosinophils were observed.

### 4.6. Immunohistochemistry

TLR2 and TLR4 protein expression in DNFB-induced skin lesions was evaluated by immunohistochemistry. Fresh skin lesions were removed, embedded in optimal cutting temperature (OCT) tissue embedding agent, and stored in liquid nitrogen or in a –80 °C freezer. The frozen sections (6 μm) were fixed in acetone at 40 °C for 10 min and washed three times for 5 min using phosphate-buffered saline (PBS). After washing, the sections were incubated with 3% hydrogen peroxide for 10 min to remove endogenous peroxidase activity. The sections were washed three times with PBS (5 min each), blocked with serum (10% normal rabbit serum for TLR2 test and 10% normal sheep serum for TLR4 test [Merck]), and incubated for 10 min at room temperature; the serum was then poured out without washing. Primary antibodies were added and incubated overnight at 4 °C. Subsequently, the sections were washed three times with PBS (5 min each) and incubated with streptomycin labeled with horseradish enzyme at 37 °C for 20 min, then washed again with PBS three times (5 min each) and stained with diaminobenzidine (DAB). Finally, the sections were rinsed with tap water, re-dyed, and sealed.

### 4.7. Enzyme-Linked Immunosorbent Assay (ELISA)

According to the instructions of each kit, the levels of serum cytokines, including IFN-γ (Th1 cell cytokine), IL-4, IL-5, IL-13, IL-31 (Th2 cell cytokines), IL-17, and IL-22 (Th17 and Th22 cell cytokines), were detected using the double antibody sandwich ELISA method; the resulting product was read in a microplate reader (Thermo Fisher, Multiskan go, Waltham, MA, USA), and their contents were quantified according to a standard curve. All experiments were performed in triplicate and repeated three times.

### 4.8. RNA Isolation and Real-Time PCR

Total RNA was extracted from dorsal skin lesions using the TRIzol (ambion, 15596026) method [31]. RNA (1 µg) was used as a template to synthesize cDNA. Reverse transcription reactions were performed using 5 mM reaction buffer, 10 mM dNTPs, 0.02 µg/µL Oligo (dT) primers, 40 U/μL RNase inhibitor, and 200 U/μL MMLV reverse transcriptase at 42 °C for 1 h. The reactions were stopped and incubated at 72 °C for 7 min, followed by storage of the cDNA products at 20 °C. RT-PCR was performed on a CFX 96 Real-Time PCR Detection System (Bio-Rad) by using the synthesized cDNA and specific primers for mice IFN-γ, IL-4, IL-5, IL-13, IL-31, IL-17, IL-22, TLR2, TLR4, and β-actin (loading control). The following primer sequences were used:

IFN-γ, F 5′-CAA CAG CAA GGC GAA AAA GGA-3′ and R 5′-GTC ACT GCA GCT CTG AAT GTT TCT T-3′; IL4, F 5′-ATT GAT GGG TCT CAA CCC CC-3′ and R 5′-TCG TTG CTG TGA GGA CGT TT-3′; IL-5, F 5′-AGA CTT CAG AGT CAT GAG AAG GA-3′ and R 5′-GTC TCT CCT CGC CAC ACT TC-3′; IL-13, F 5′-ACA TCA GAC TTT GCT GGG GAG-3′ and R 5′-TAC AGT GAG GTA GCA GAG TTA TG-3′; Il-31, F 5′-CCA GGC TCC AGA GAC CAC AGG-3′ and R 5′-CAC CAG GGT AGG CTT CGT TGT TC-3′; IL-17, F 5′-TGA TGC TGT TGC TGC TGC TGA G-3′ and R 5′- CAC ATT CTG GAG GAA GTC CTT GGC-3′; IL-22, F 5′-TGC GAT CTC TGA TGG CTG TC-3′ and R 5′-CCT CGG AAC AGT TTC TCC CC-3′; TLR2 F 5′-GGC AGC TTT CTC TTT AGG ACA-3′ and R 5′-GAT TGC GGA CAC ATC TCC TG-3′; and TLR4, F 5′-CAC AGA AGA GGC AAG GCG ACA G -3′ and R 5′-GAA TGA CCC TGA CTG GCA CTA ACC-3′.

### 4.9. Statistical Analysis

Data are presented as mean ± standard error of mean (SEM). Statistical analysis was performed using SPSS 20.0 (Chicago, IL, USA). The levels of multiple cytokines and the expression of TLR2 and TLR4 were analyzed using one-way analysis of variance. Statistical significance was set at *p* < 0.05.

## Figures and Tables

**Figure 1 ijms-23-11371-f001:**
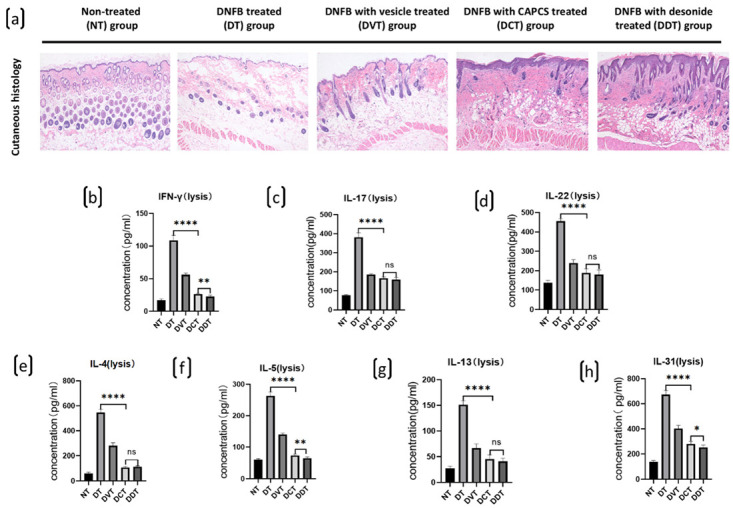
Cutaneous histology and expression of inflammatory cytokines of dorsal skin lesion of in each group of BALB/c mice. (**a**) Cutaneous histology of mice in each group: no hyperplasia or hyperkeratosis of epidermis, no infiltration of inflammatory cells was found in NT group; hyperkeratosis, dyskeratosis, and obvious inflammatory cell infiltration were found in DT and DVT groups; less inflammatory cell infiltration was shown in DCT and DDT groups (200×). (**b**–**h**) The expressions of inflammatory cytokines, including IFN-γ, IL-4, IL-5, IL-13, IL-31, IL-17, and IL-22, were increased significantly in DNFB-induced AD-like dorsal skin lesions in the DT, DVT, DCT, and DDT groups, in comparison to normal dorsal skin in NT group, and the levels of the cytokines expression were significantly decreased in the DCT and DDT groups, in comparison to the DT and DVT groups (data are presented as the mean ± S.E.M; ns: no significance, * *p* < 0.05, ** *p* < 0.01, and **** *p* < 0.0001).

**Figure 2 ijms-23-11371-f002:**
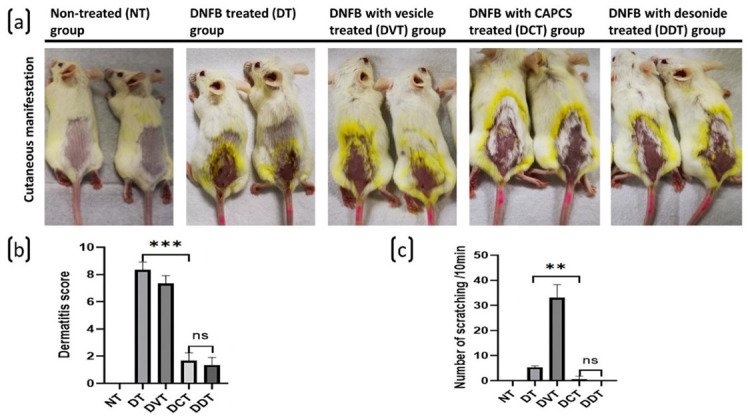
Cutaneous manifestations of DNFB-induced AD-like symptom, dermatitis score, and scratching behavior in BALB/c mice. (**a**) Cutaneous manifestations of mice in each group. No skin lesions of AD-like symptom were seen in the NT group, obvious erythema, oozing, edema, excoriation, and crust were observed in DT and DVT groups, and noticeable erythema and dryness, mild edema and crust, without obvious oozing, excoriation, or lichenification were found in the DCT and DDT groups. (**b**) The final dermatitis scores were 0 in the NT group, significantly lower in the DCT and DDT groups than the DT and DVT groups, and the scores were similar between the DCT and DDT groups. (**c**) Scratching behavior was observed every 3 days before and after the successful modeling, and the scratching times of mice in 10 min were recorded and counted by video camera. The scratching behavior of mice was obviously reduced in the DCT and DDT groups, and there were no significant differences observed between these two groups (data are presented as the mean ± S.E.M; ns: no significance, ** *p* < 0.01 and *** *p* < 0.001).

**Figure 3 ijms-23-11371-f003:**
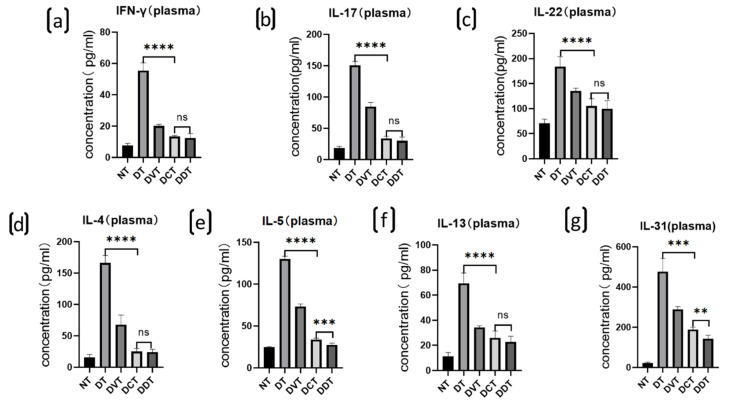
Expression levels of the plasmatic inflammatory cytokines in each group of BALB/c mice. (**a**–**d**,**f**) The expression levels of multiple cytokines, including IFN-γ, IL-17, IL-22, IL-4, and IL-13, were increased significantly in the DT, DVT, DCT, and DDT groups in comparison to the NT group, the levels of these cytokines were significantly decreased in the DCT and DDT groups in comparison to the DT and DVT groups, and there was no significance between the DCT and DDT groups; (**e**,**g**) The expression levels of IL-5 and IL-31 were increased significantly in the DT, DVT, DCT, and DDT groups in comparison to the NT group, the levels were significantly decreased in the DCT and DDT groups when compared with the DT and DVT groups, and these levels were significantly higher in DCT group than DDT group (data are presented as the mean ± S.E.M; ns: no significance, ** *p* < 0.01,*** *p* < 0.001, and **** *p* < 0.0001).

**Figure 4 ijms-23-11371-f004:**
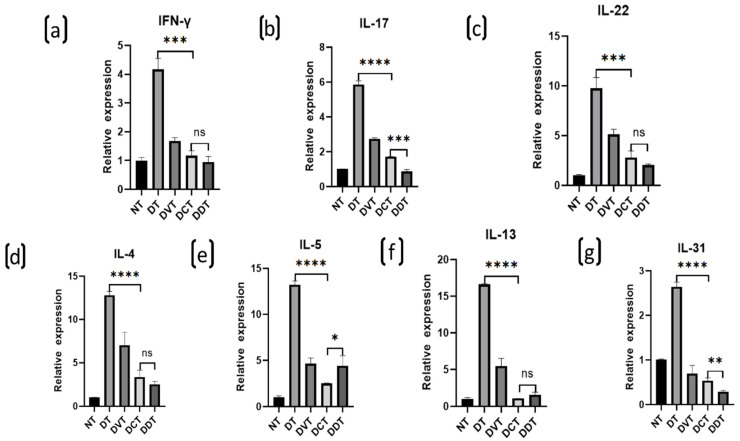
Expression levels of the inflammatory cytokines of dorsal skin lesions determined by qRT-PCR in each group of BALB/c mice. (**a**,**c**,**d**,**f**) The expression levels of IFN-γ, IL-22, IL-4, and IL-13 were increased significantly in the DT, DVT, DCT, and DDT groups in comparison to the NT group, the levels were significantly decreased in the DCT and DDT groups in comparison to the DT and DVT groups, and there was no significance between the DCT and DDT groups; (**b**,**g**) The expression levels of IL-17 and IL-31 were increased significantly in the DT, DVT, DCT, and DDT groups in comparison to the NT group, the levels were significantly decreased in the DCT and DDT groups when compared with the DT and DVT groups, and the levels were significantly higher in DCT group than DDT group; (**e**)The expression level of IL-5 was increased significantly in the DT, DVT, DCT, and DDT groups in comparison to the NT group, the level was significantly decreased in the DCT and DDT groups when compared with the DT and DVT groups, and this level was significantly lower in DCT group than DDT group (data are presented as the mean ± S.E.M; ns: no significance, * *p* < 0.05, ** *p* < 0.01, *** *p* < 0.001, and **** *p* < 0.0001).

**Figure 5 ijms-23-11371-f005:**
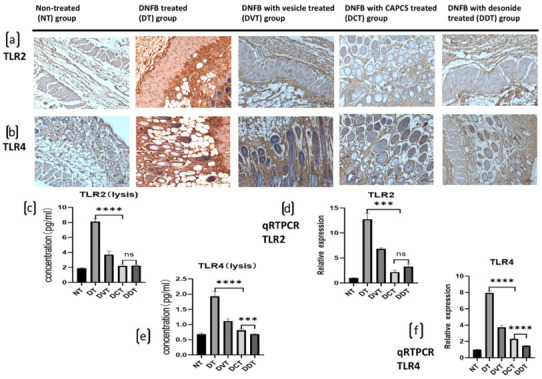
TLR2 and TLR4 immunohistochemical staining, protein expression in dorsal skin lesions, and qRT-PCR in each group of BALB/c mice. (**a**) Immunohistochemical staining of TLR2 in dorsal AD-like skin lesion (200×). (**b**) Immunohistochemical staining of TLR4 in dorsal AD-like skin lesion. (**c**,**d**) Expression of TLR2 was significantly increased in the DT, DVT, DCT, and DDT groups, in comparison to the NT group, and the expression was significantly decreased in the DCT and DDT groups, in comparison to the DT and DVT groups in skin lesions. (**e**,**f**) Expression of TLR4 was significantly increased in the DT, DVT, DCT, and DDT groups, in comparison to the NT group, and the expression was significantly decreased in the DCT and DDT groups, in comparison to the DT and DVT groups in skin lesions (data are presented as the mean ± S.E.M; ns: no significance, *** *p* < 0.001, and **** *p* < 0.0001).

**Figure 6 ijms-23-11371-f006:**
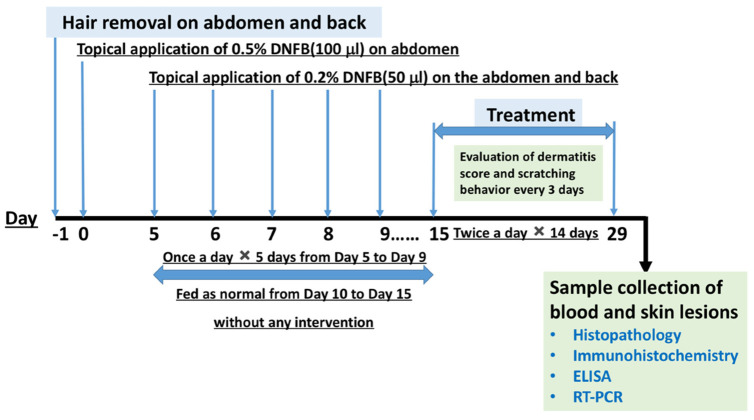
Experimental schedule for establishment of AD-like symptom model and treatment plan in BALB/c mice.

## Data Availability

All raw data included in this study could be available upon request by contacting the corresponding author J.L. via email: jingyili@wchscu.cn.

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
