# Peer review of "Calcium-Based Antimicrobial Peptide Compounds Attenuate DNFB-Induced Atopic Dermatitis-Like Skin Lesions via Th-Cells in BALB/c Mice"

_ijms, 2022, doi:10.3390/ijms231911371_

Round 1

Reviewer 1 Report

The article is interesting, but a bit underdeveloped. In my opinion, it should be sent to the authors for correction. The article may have potential in view of the increasing prevalence of AD.

Particularly, the section of materials and methods needs to be improved to ensure reproducible results. The Data Availability Statement should also be improved, since the study obtained raw results.

MAJOR COMMENTS:

- information on the prevalence of AD should be completed

- quality of figure 2b and 2c should be improved

- source of dinitrofluorobenzene, CAPCS and desonide should be mentioned

- product numbers of kits should be added to allow reproducible results

- information on the equipment used, e.g. spectrophotometer, should be supplemented

- information on blood collection should be completed

- line 335 – should be “did not” instead of “didn’t”

- line 336 – “Staphylococcus aureus” should be an abbreviated name in italics

- is anything known about the toxicity of CAPCS?

- the results are not fully discussed – discussion should be expanded

- references should be adjusted to the Journal's requirements

Reviewer 2 Report

The authors studied how calcium-based antimicrobial peptides can reduce skin lesions caused by dinitrofluorobenzene. They have a rational design for their experiment, including several contrast groups, with no treatment or other known treatment. The result also supports their conclusion that calcium-based antimicrobial peptides have a similar effect as desonide, which they used as a comparison group. 

I want to address some small things that can be improved. 

(1) I would like to have one more contrast group, whist uses calcium-based antimicrobial peptides on a non-treated group, and one more comparison group, which uses both desonide and calcium-based antimicrobial peptides. Calcium-based antimicrobial peptides on no DNFB treated skin would help to understand what calcium-based antimicrobial peptides do to healthy skin tissue. The cocktail with CAPCS and desonide treatment will provide more information on whether the two drugs are under similar mechanisms can help enhance the cure. 

(2) The result supported that CAPCS can have a therapeutic effect. However, the mechanism has not been surveyed experimentally, even if the authors discussed it a lot.  

(3) The authors also realized some limitations of this study. I believe a bacteriological experiment is needed to demonstrate the antimicrobial activity of CAPCS. 

Round 2

Reviewer 1 Report

All comments were addressed. Article can be published. 

Reviewer 2 Report

I recommend publishing this revised manuscript.